# Hypertension in Patients with Insulin Resistance: Etiopathogenesis and Management in Children

**DOI:** 10.3390/ijms23105814

**Published:** 2022-05-22

**Authors:** Veronica Maria Tagi, Francesca Mainieri, Francesco Chiarelli

**Affiliations:** Department of Pediatrics, University of Chieti, Via dei Vestini 5, 66100 Chieti, Italy; veronica.tagi@gmail.com (V.M.T.); mainieri.francesca@gmail.com (F.M.)

**Keywords:** hypertension, insulin resistance, diabetes mellitus, children

## Abstract

Insulin resistance (IR) is a key component in the etiopathogenesis of hypertension (HS) in patients with diabetes mellitus (DM). Several pathways have been found to be involved in this mechanism in recent literature. For the above-mentioned reasons, treatment of HS should be specifically addressed in patients affected by DM. Two relevant recently published guidelines have stressed this concept, giving specific advice in the treatment of HS in children belonging to this group: the European Society of HS guidelines for the management of high blood pressure in children and adolescents and the American Academy of Pediatrics Clinical Practice Guideline for Screening and Management of High Blood Pressure in Children and Adolescents. Our aim is to summarize the main pathophysiological mechanisms through which IR causes HS and to highlight the specific principles of treatment of HS for children with DM.

## 1. Introduction

IR is the principal mechanism responsible for the development of HS in patients with DM [1]. In fact, it contributes to an increase blood pressure (BP) in several ways, including the enhanced tissue angiotensin II (AngII) and aldosterone activities [2,3], the increased sympathetic nervous system activity [4], and oxidative stress [5]. Nevertheless, the hypothesis of the “endothelial IR” has been postulated, according to which endotelial disfunction precedes peripheric IR due to the impairment of blood flow in peripheral tissues. This mechanism is mediated by increased oxidative stress, through a protein kinase C β-dependent pathway. In fact, the suppression of reactive oxygen species-dependent pathways in the endothelium has been shown to restore insulin delivery to peripheral organs by preserving nitric oxide (NO) availability [6,7].

The higher risk of developing cardiovascular morbidity in children with a youth-onset Type 2 DM (T2D) is well known. Longitudinal data from the Treatment Options for T2D in Adolescents and Youth (TODAY) study revealed that in a group of 677 participants with a mean age of 14 ± 2 years the cumulative incidence of HS, LDL-C dyslipidemia, and hypertriglyceridemia was 59%, 33%, and 37% respectively and at the end of a mean 10.2 ± 4.5 years follow-up 54% had ≥2 cardiovascular risk factors in addition to T2D. Male sex, non-Hispanic white race/ethnicity, obesity, poor glycemic control, lower insulin sensitivity, and reduced β-cell function were identified as the main risk factors [8].

Agbaje et al. investigated the temporal causal longitudinal associations of carotid-femoral pulse wave velocity (cfPWV), an index of arterial stiffness, with the risk of developing IR, measured with the homeostatic model assessment (HOMA) index, in a group of 3862 followed-up 17.7-year-old participants from the Avon Longitudinal Study of Parents and Children. HOMA index is a paradigm model which allows for determining the IR rate using only the fasting glucose and insulin values. Levels above 2.9 signal significant insulin resistance. A higher value of HOMAIR corresponds to a more severe IR. The study showed that a higher cfPWV at 17.7 years was associated with higher HOMA index at age 24.5 years, supporting the hypothesis according to which arterial stiffness in adolescence may be a causal risk factor for hyperinsulinemia and IR in young adulthood [9].

Adults with DM are two to four times more likely to die from heart disease than those who do not have DM [10].

The prevalence of HS is higher in patients with DM than in the general population. In fact, it is estimated that about 69.0% have a systolic blood pressure of 140 mmHg or higher or diastolic blood pressure of 90 mmHg or higher or are on prescription medication for their high blood pressure [11]. If HS and DM are both present, the incidences of cardiovascular diseases (CVD) and mortality increase, such as the risks of nephropathy and retinopathy [12,13,14].

Given these premises, specific management of HS in children affected by DM is mandatory. The two most relevant societies that have recently proposed a peculiar treatment and follow-up of children with DM in their guidelines are the European Society of HS [15] and the American Academy of Pediatrics [16]. In this review, we will provide an overview of the most recent knowledge about the etiopathogenesis of the development of HS in patients with IR and of the principal recommendations for the treatment of HS in children with DM. 

## 2. Etiopathogenesis

The association between IR and HS is multifactorial [1].

IR is involved in the development of HS and atherosclerotic cardiovascular disease through three main mechanisms [17]: the basic molecular etiology of IR [18,19,20,21,22,23,24,25,26,27], the compensatory hyperinsulinemia that occurs in response to IR [24,28,29,30,31,32,33,34], and the association between IR and some cardiometabolic abnormalities [28,29,30,35,36].

Insulin acts on its target organs phosphorylating a transmembrane-spanning tyrosine kinase receptor, the insulin receptor (IR). It binds to the α subunit of its receptor activating the tyrosine kinase of the β subunit of the receptor, causing autophosphorylation and phosphorylation of several IR substrates (IRS), such as IRS-1 and IRS-2 [37]. These substrates activate phosphatidylinositol 3-kinase (PI3K), which stimulates Akt, a serine/theronine kinase, which in turn stimulates the glucose uptake through the translocation of GLUT-4, the major glucose transporter to the cell membrane [38]. Akt is also responsible for the activation of nitric oxide synthase, which leads to the production of NO. If this pathway is impaired, NO, which is a potent vasodilator and antiatherogenic agent [28,29], is not produced, favouring vascular resistance, HS and atherogenesis (Figure 1).

One of the mechanisms through which hyperinsulinemia causes an increase of BP is dysregulation of peripheral vascular resistance, stimulating the sympathetic system and therefore causing vasoconstriction [4]; furthermore, hyperinsulinemia contributes to the development of HS-related target organ damage, in particular: impairment of cell membrane ion exchange, enhanced sympathetic nervous and renin-angiotensin systems, suppressed atrial natriuretic peptide activities, sodium retention, and plasma volume expansion lead to chronic kidney disease, left ventricular hypertrophy and carotid atherosclerosis [1].

Moreover, insulin-resistant states, such as T2D, are associated with several cardiometabolic abnormalities factor abnormalities, including elevated PAI-1, increased fibrinogen, and higher platelet stickiness, which are important cardiovascular risk factors [39,40].

Obesity is one of the leading causes of IR [14,28,29], and is strongly associated to atherosclerotic cardiovascular disease [41,42,43].

IR in obese subjects is mainly determined by lipotoxicity: the effect of excessive lipid accumulation which occurs when energy intake exceeds energy consumption [28,29,30]. Elevated free fatty acids in blood promote lipid deposition in tissues, including vessels [44,45,46,47,48,49] and activation of inflammatory pathways [43]. Excessive fat in adipocytes induces enhances the secretion of proinflammatory/prothrombotic cytokines, such as TNFa, PAI-1 and resistin, which promote atherogenesis [50,51,52,53].

## 3. Management of HS in Children with DM

According to the SEARCH for DM in Youth Study, only 7.4% of youth with type 1 DM (T1DM) and 31.9% of youth with T2D who attended a visit between 2001 and 2010 knew their BP status and, after becoming aware of the diagnosis of HS, only 57.1% of patients with T1DM and 40.6% of patients with T2DM achieved good BP control [54]. In comparison with youth with DM1, those with DM2 were older and more likely women, had an older age at diagnosis, were more likely to be obese (26.5 vs. 11.1%) and belong to ethnic minorities (19–76% vs. 2.9%). Furthermore, they presented higher systolic and diastolic BP, lower glycated hemoglobin, and higher prevalence of microalbuminuria [47]. Recent data confirm that HS is underdiagnosed in youths with DM1 younger than 13 years of age [55].

The European Society of Hypertension (ESH) and the American Academy of Pediatrics (AAP) have recently published their new guidelines for the treatment of elevated BP in children. In both the documents, specific indications are reserved for the detection and management of HS in children with DM [15,16].

### 3.1. Definition

The definitions of elevated BP and HS in children provided by the ESH and the AAP present some differences [15,16]. First, they suggest two distinct age classifications: according to the ESH BP percentiles should be considered until the age of 15 years, while for the AAP they should be used only until the age of 12 years. Second, the ESH calls the transient stage between normal and HS high-normal BP, while the AAP talks about elevated BP. Third, the AAP’s values for the suggested definitions substantially differ from the ESH ones, generally being lower. Table 1 illustrates the principal differences in the two societies’ definitions.

### 3.2. Prevention 

Data about the efficacy of preventive antihypertensive treatment in children with DM is still lacking; however, evidence in adults suggests that a strict BP control should be aimed [15].

Furthermore, in obese adults, ACE inhibitors and AngII Receptor Blockers (ARB) seem to reduce the incidence of new-onset DM and may increase insulin sensitivity [56].

### 3.3. Lifestyle Intervention 

Lifestyle intervention is recommended in children affected by DM. 

The National Heart Lung and Blood Institute’s Growth and Health Study demonstrated that consuming ≥2 servings of dairy and ≥3 servings of fruit and vegetables daily was associated with lower BP in childhood not only in healthy subjects but also in children and adolescents with DM [16,57]. Furthermore, an improved diet was shown to be related to a lower BP in adolescents with elevated BP [58] and youth with T2DM [59]. 

Exercise is fundamental as well, given its effects on both lowering BP and improving insulin sensitivity [15,16].

Other lifestyle changes, such as a reduction in salt intake, tobacco cessation, and appropriate sleep hygiene, are known to improve metabolic health and reduce BP in patients with DM [10].

### 3.4. First-Line Pharmacologic Treatment 

Although lifestyle intervention plays an important role in lowering BP in youth with IR, pharmacological interventions are required to achieve optimal BP [10].

Children at any stage of HS with DM should undergo pharmacologic treatment starting with a single medication at the low end of the dosing range. Depending on repeated BP measurements, the dose of the initial medication can be increased every 2 to 4 weeks until BP is controlled, the maximal dose is reached, or adverse effects occur [16]. 

According to the American DM Association (ADA), a drug class with demonstrated cardiovascular benefits such as a renin-angiotensin system (RAS) inhibitor (angiotensinconverting enzyme inhibitor [ACEI] or ARB) should be chosen as first-line therapy, unless there is an absolute contraindication [15,16]. These two families of drugs are useful also in case of microalbuminuria, because of their antiproteinuric effect [15,60]. 

It has been demonstrated that the ACEI do not only act as lowering agents, but also as modulators of metabolic anomalies [61]. Several mechanisms of function are involved in the treatment of the latter pathological condition. First, ACEI interfere with the conversion of angiotensin I into AngII; furthermore, they increase the circulating level of the bradykinin by inhibiting the kininase II breakdown [62] and higher kinin levels are responsible for a major production of prostaglandins (prostaglandin E1) and NO, which improve exercise-induced glucose metabolism and muscle sensitivity to insulin [63,64], therefore insulin-mediated glucose uptake increased. Moreover, ACEI have the ability of causing peripheral vasodilation, leading to an increased surface area for glucose exchange between vessels and skeletal muscles [61]. A study conducted by Morel et al. [65] demonstrated that IR was reduced in patients with high cardiovascular risk treated with enalapril for 12 weeks. Captopril was shown to have similar effectiveness in improving insulin sensitivity [66]. 

The effect of ACEI on IR may be also explained by the regulation of adipocyte cell cycle. This hypothesis is supported by data showing that higher levels of AngII interfere with the differentiation of pre-adipocytes into mature adipocytes, and therefore that fat cells are not able to store fat. Consequently, fats are stored into the liver, skeletal muscle, and pancreas, increasing IR. On the contrary, ACEI reduces AngII levels, therefore promoting the differentiation of pre-adipocytes into mature adipocytes and enhancing the storage of fat into mature adipocytes [67]. 

Another mechanism underlying the role of ACEI in modulating metabolic anomalies involves pancreatic β cell protection. In fact, reduced AngII levels leads to a milder vasoconstriction increases the islet blood flow [68], and therefore enhancing insulin releasing by β cells. 

There is no compelling evidence in favour of one drug class over another except for data supporting the early use of RAS inhibitors in patients with overt proteinuria (urine albumin-to-creatinine ratio >300 mg/g) [10]. Other antihypertensive medications (e.g., α-blockers, β-blockers, combination α- and β-blockers, centrally acting agents, potassium-sparing diuretics, and direct vasodilators) should be used only in case of no response to 2 or more of the preferred agents [15,16]. If β-blockers are a compelling indication, those with vasodilating capacity should be used [15]. 

According to the Eight Joint National Committee (JNC 8), non-black patients with DM should benefit similarly from treatment with ACEI/ARB, CCB, or thiazide-like diuretic, while black patients with DM benefit more from treatment with tha iazide-like diuretic or a CCB [69].

According to the ESH and the European Society of Cardiology (ESC), all classes of antihypertensive agents are recommended and can be used in patients with DM, keeping in mind that RAS blockers may be preferred in the presence of proteinuria or microalbuminuria [70]. 

Table 2 lists the age and dosing recommendations, contraindications, and adverse drug reactions of anti-hypertensive drugs for children. 

Once the appropriate drug has been chosen, treatment should start with the lowest recommended dose [15,71], this dose should be up-titrated until the BP falls within the target range or until the maximum recommended dose is reached or side effects are developed [15,16]; for example, in children treated with ACE inhibitors and ARB kidney function and potassium balance should be strictly monitored [15].

### 3.5. Second Agent Treatment 

If a single agent is not sufficient, a second agent can be added and up-titrated with the same modalities of the initial drug. Since salt and water retention usually occurs with many antihypertensive medications, a thiazide diuretic is often the preferred second agent [15,16]. The preferred combinations are a thiazide diuretic with an ARB, a calcium antagonist, or an ACE-inhibitor. The association between thiazides and β-blockers increases the risk of developing DM and should therefore be avoided, especially in children with IR or with other risk factors for T2DM [72].

The association of an ACE inhibitor and an ARB is always contraindicated due to the absence of demonstrated benefits in front of an increase in adverse events.

Few drug-related side effects may be dose-limiting. In these cases, the early addition of a second agent or complete replacement of the initial agent are both therapeutic options [15].

### 3.6. BP Goal

In adults, there is no clear evidence that BP should be lower in diabetic than in nondiabetic hypertensive patients [73]. In children, data are lacking; however, it has been observed that youth with DM1 or DM2 develop early atherosclerotic lesions before the age of 30 [74].

Since evidence has emerged that markers of target organ damage, such as increased LVMI, can be detected among some children with BP in the >90th percentile (or >120/80 mmHg) but in the <95th percentile [75,76,77], the American Heart Association suggests that the BP goal should be lower than the 90th percentile for age, sex and height, or below 130/80 mmHg at age 16 and above [78], and the ESH and the AAP provide the same recommendations [15,16]. 

A post-hoc analysis of a trial on DM1 suggests that a lower BP target may be beneficial in reducing urinary albumin excretion and the risk of developing proteinuria [73].

### 3.7. Heart Failure 

Children and adolescents with DM have an increased risk of developing early cardiovascular events and heart failure than their non-diabetic peers [74]. Given the high rates of coronary artery disease among patients with DM, patients with DM have a higher risk of developing ischemic cardiomyopathy and, consequently, heart failure; however, heart failure and IR are not only linked by a higher prevalence of ischemic heart disease since heart failure is more frequent in patients with DM despite the presence of ischemic disease [75].

In case of heart failure, first-line therapy includes salt restriction and agents that target the renin–angiotensin–aldosterone-system combined with β-blockers in low doses if necessary to achieve BP control. Adding diuretics is recommended in patients with volume overload [75,76]. In the case of coarctation of the aorta, presenting with heart failure and upper limb HS, surgery or catheter intervention is the first treatment of choice [15]. 

Treatment of a hypertensive crisis causing acute heart failure includes the use of intravenous vasodilatory agents (nicardipine), with nitroprusside limited to situations where other agents fail, or to brief periods of time [73].

### 3.8. Metabolic Syndrome 

IR and HS are part of a cluster of cardiometabolic risk factors named metabolic syndrome (MetS). Hypertension is often associated with other risk factors for IR present in metabolic syndrome, particularly type 2 diabetes, in both adults and children. In adults, through randomized multicenter studies, it has recently been possible to demonstrate the importance of intensified multifactorial treatment to reduce morbidity in such situations. It would be desirable that RCTs were designed to evaluate the long-term clinical impact of controlling the risk factors associated with IR also in the pediatric population [77].

Even though non-alcoholic fatty liver disease (NAFLD) is not yet included in the current definition of MetS, it is well known to be its frequent hepatic manifestation. MetS and NAFLD are two different entities sharing common clinical and physio-pathological features, with insulin resistance as the most relevant. The increased MS incidence worldwide, above all due to changes in diet and lifestyle, is associated with an equally significant increase in NAFLD, in children too [79]. 

Despite the definition of MetS in children and adolescents being still uncertain, what is clear is that a combination of physical exercise and dietary intervention, together with pharmacologic treatment and bariatric surgery in selected cases, in order to reduce overweight and obesity are crucial for an improvement of all the clinical conditions [80]. The use of metformin for abdominal obesity seems to be effective in the short-term but information about the efficacy over time is lacking [78]; obviously, the rest of the cardiometabolic risk factors should be treated singularly as well [15]. 

### 3.9. Glucose Lowering Therapies

The role of some glucose-lowering molecules has been evaluated also in the management of HS. For example, sodium-glucose co-transporter-2 (SGLT2) inhibitors are used in adult patients with T2DM; however, their use in the pediatric population has been proposed by many experts [81]. Recent evidence suggests that inhibiting SGLT2 may have an effect on lowering BP and reducing body weight, significantly reducing cardiovascular risk [82]. 

SGLT2 inhibitors block the renal sodium-glucose cotransporter-2, increasing the urinary excretion of glucose, and leading to osmotic diuresis [75]. Luo et al. [82] conducted a study on mice with HS and DM, with the aim of finding whet (sEH) inhibitor 1-trifluoromethoxyphenyl3-(1-propionylpiperidin-4-yl) urea (TPPU) could reduce their BP and glucose. sEH is ubiquitary and highly expressed in the human liver, followed by the kidney [83,84]; it hydrolyzes epoxyeicosatrienoic acids (EETs) [85], which are metabolites of arachidonic acid produced by the cytochrome P450 pathway [86]. Data from this study demonstrated that TPPU administration ameliorated HS and IR by decreasing renal glucose and sodium reabsorption via inhibitory kappa B kinase α/β (IKKα/β)/NF-κB signalling pathway-mediated SGLT2 inhibition [82].

Regarding heart failure, no relationships between the degree of glycated hemoglobin (HbA1c) reduction under glucose-lowering treatment and reduction in the incidence of heart failure have been observed [87]. Data from the Candesartan in Heart failure: Assessment of Reduction in Mortality and Morbidity (CHARM) program, showed that an increase in HbA1C level of 1% was associated with a 25% increased risk of cardiovascular (CV) death, hospitalization for worsening heart failure, and all-cause death [88]. Findings from a Veterans Affair registry study on more than 5000 outpatients with heart failure and DM, supported the hypothesis that patients who achieved moderate glucose control (HbA1C >7.1% and <7.8%) had the lowest mortality [89].

Even though intensive glucose control has been demonstrated to reduce microvascular complications of DM [90], a strict glucose control does not seem to strongly impact the reduction in CV mortality among patients with DM [91]. Furthermore, some specific medications seem to worsen outcomes among patients with DM and heart failure and have therefore required caution when used in this population; however, other molecules, such as SGLT-2, have been demonstrated to have some CV benefits. The multicenter Empagliflozin, Cardiovascular Outcomes, and Mortality in T2D trial (EMPA-REG OUTCOME) has demonstrated positive effects of SGLT-2 inhibitors on CV outcomes [92]. A randomized case-control study was conducted on the treatment with empagliflozin 10 or 25 mg in more than 7000 subjects with a diagnosis of atherosclerosis. Empagliflozin was demonstrated to statistically significantly reduce the incidence of CV mortality, nonfatal MI, or nonfatal stroke; a significant reduction in heart failure incidence was also observed. In the CANagliflozin cardioVascular Assessment Study (CANVAS), [93] patients with T2DM and high CV risk (including those with and without established CV disease) randomly received, treated with canagliflozin or placebo and were followed for more than 3.5 years, revealing that canagliflozin significantly reduced the incidence of death from CV causes, nonfatal MI, or nonfatal stroke; moreover, the secondary outcome, which was the reduction of hospitalisation for heart failure, was reached in patients treated with canagliflozin. Dapagliflozin has been shown to have similar effects on BP and heart failure hospitalisation in a trial with a predominant observation of patients without established CV disease and in observational studies [93,94]. Although CV risk has a major impact in African-American patients with IR, a concerning representation of this population is present in the above-mentioned studies [95]. Hitherto, only one randomised trial is present in the literature, focusing on minorities and analysing 150 African-Americans with DM and HS randomised to empaglilozin or placebo. Empagliflozin was shown to reduce blood pressure, body weight, and HbA1C [96]. Afterwards, these findings were confirmed by a large real-world population study of African-American patients with DM and HS from the DM Collaborative Registry (an observational US registry of patients recruited from primary care, cardiology, and endocrinology practices). Among the 74,290 African-American patients with DM in this registry, 34.5% had a systolic BP greater than or equal to 140 mm Hg; of these, only 1.7% had been prescribed an SGLT-2 inhibitor. Assuming an 8 mmHg reduction in BP with the use of these medications, the mean estimated 5-year risk of CV death was estimated to be reduced from 7.7% to 6.2% if SGLT-2 inhibitors were used in this population [97]. Interestingly, SGLT-2 have been demonstrated to reduce CV events mainly through their effect on obesity and BP rather than their impact on blood glucose levels [95,96]. The mechanism of action, underlying the effect of SGLT-2 inhibitors in decreasing BP is not fully understood; however a multifactorial action has been postulated including osmotic and natriuretic diuresis and decrease in arterial stiffness and sympathetic tone [97]; their mechanism of action supports the hypothesis according to which reduction of risk of heart failure is higher than the atherothrombotic-related effects [98].

### 3.10. Follow-Up

The AAP recommends that all children ≥3 years of age, especially those with DM, should have BP measured at every health encounter [16,99,100,101,102].

According to the ESH, in children with DM, regular ABPMs at 6–12-month intervals should be performed to rule out selective nocturnal HS [15].

## 4. Conclusions

Consistent evidence suggests that IR is involved in the development of HS and atherosclerotic cardiovascular disease.

Children with T1DM and T2DM and HS deserve particular attention in the diagnosis and specific management. 

Besides lifestyle intervention, children at any stage of HS with DM should undergo pharmacologic treatment starting with ACE inhibitors or ARB and eventually adding a second drug if BP goal <90th percentile for age, sex and height, or below 130/80 mmHg at age 16 and above is not reached.

Strict follow-up, in order to early detect long-term complications of HS, is particularly recommended in this category of patients.

Glucose-lowering drugs, especially SGLT-2 inhibitors, have been evaluated as potential anti-hypertensive agents; however, their use in the pediatric population has not been approved yet; therefore, large-population clinical trials should be started in order to make this therapeutic option available for children and adolescents.

## Figures and Tables

**Figure 1 ijms-23-05814-f001:**
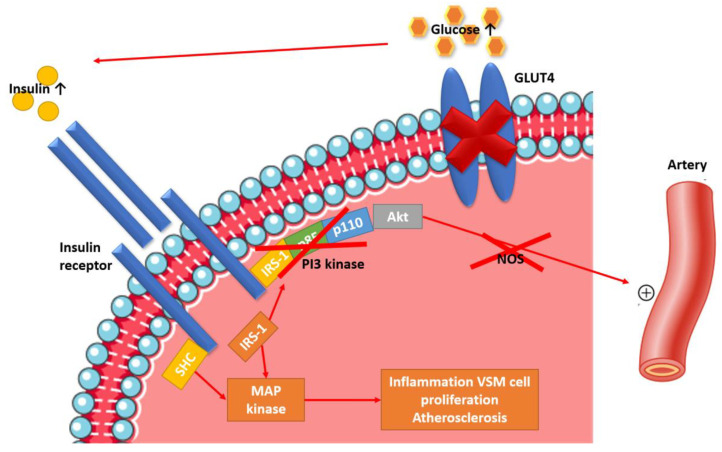
Pathophysiological mechanisms linking IR and hypertension in children with DM. The signaling is impaired at the level of IRS-1, therefore, glucose transport is decreased and nitric oxide synthase activation is impaired, leading to decreased NOS production and therefore vasoconstriction. At the same time, insulin signalling through the MAPK pathway remains normally sensitive to insulin. For this reason, compensatory hyperinsulinemia (secondary to insulin resistance in the IRS-1/PI3K pathway) causes excessive stimulation of the MAPK pathway, which is involved in inflammation, vascular smooth muscle cell proliferation, and atherogenesis. Ab—Abbreviations: NOS—nitric oxide synthase; SHC—Src homology collagen.

**Table 1 ijms-23-05814-t001:** Comparison between the ESH’s and the AAP’s definitions of elevated BP and HS in children.

	ESH		AAP	
Category	0–15 years SBP and/or DBP (percentile values)	≥16 years SBP and/or DBP (mmHg)	1–12 years (percentile values)	≥13 years (mmHg)
Normal	<90th	<130/85	<90th	<120/<80
High-normal/Elevated BP	≥90th to <95th	130 to 139/85 to 89	≥90th to <95th	120/<80 to 129/<80
Stage 1 HS	95th to 99th +5 mmHg	140 to 159/90 to 99	95th to 95th +12 mmHg	130/80 to 139/89
Stage 2 HS	>99th + 5 mmHg	160 to 179/100 to 109	≥95th + 5 mmHg	≥140/90

Abbreviations: ESH—European Society of Hypertension; AAP—American Academy of Pediatrics; HS—hypertension; BP—blood pressure; SBP—systolic blood pressure; DBP—diastolic blood pressure.

**Table 2 ijms-23-05814-t002:** Antihypertensive drugs: age and dosing recommendations, contraindications, and adverse drug reactions.

Drug	Age	Initial Dose (mg/kg per Dose)	Maximal Dose (mg/kg per Dose)	Dosing Interval	Contraindications and Adverse Drug Reactions
**ACE inhibitors**					**Contraindications:** pregnancy, angioedema. **Common ADR:** cough, headache, dizziness, asthenia. **Severe ADR:** hyperkaliemia, acute kidney injury, angioedema, fetal toxicity
Benazepril	≥6 y	0.2 (up to 10 mg/d)	0.6 (up to 40 mg/d)	1/day	
Captopril	Infants	0.05	6	1–4/day	
	Children	0.5	6	3/day	
Enalapril	≥1 mo	0.08 (up to 10 mg/d)	0.6 (up to 40 mg/d)	1–2/day	
Fosinopril	≥6 y	0.1 (up to 5 mg/d)	40 mg/d	1/day	
	˂50 kg				
	≥50 kg	5 mg/d	40 mg per d		
Lisinopril	≥6 y	0.07 (up to 10 mg/d)	0.6 (up to 40 mg/d)	1/day	
Ramipril	-	1.6 mg/m^2^/d	6 mg/m^2^/d	1/day	
Quinapril	-	5 mg/d	80 mg/d	1/day	
**ARB**					**Contraindications:** pregnancy. **Common ADR:** headache, dizziness. **Severe ADR:** hyperkaliemia, acute kidney injury, fetal toxicity
Candesartan	1–5 y	0.02 (up to 4 mg/d)	0.4 (up to 16 mg/d)	1–2/day	
	≥6 y				
	˂50 kg	4 mg/d	16 mg/d		
	≥50 kg	8 mg/d	32 mg/d		
Irbesartan	6–12 y	75 mg/d	150 mg/d	1/day	
	≥13 y	150 mg/d	300 mg/d		
Losartan	≥6 y	0.7 (up to 50 mg/d)	1.4 (up to 50 mg/d)	1/day	
Olmesartan	≥6 y			1/day	
	˂35 kg	10 mg	20 mg		
	≥35 kg	20 mg	40 mg		
Valsartan	≥6 y	1.3 (up to 40 mg/d)	2.7 (up to 160 mg/d)	1/day	
**Thiazide diuretics**					**Contraindications:** anuria. **Common ADR:** hypokalemia, dizziness. **Severe ADR:** dysrhythmias, cholestatic jaundice, new-onset DM, pancreatitis.
Chlorthalidone	Child	0.3	2 (up to 50 mg/d)	1/day	
Chlorothiazide	Child	10	20 (up to 375 mg/d)	1–2/day	
Hydrochlorothiazide	Child	1	2 (up to 37.5 mg/d)	1–2/day	
**Calcium channel blockers**					**Contraindications:** hypersensitivity to CCB. **Common ADR:** flushing, peripheral edema, dizziness. **Severe ADR:** angioedema.
Amlodipine	1–5 y	0.1	0.6 (up to 5 mg/d)	1/day	
	≥6 y	2.5 mg	10 mg		
Felodipine	≥6 y	2.5 mg	10 mg	1/day	
Isradipine	Child	0.05–0.1	0.6 (up to 10 mg/d)	Capsule: 2–3/day Extended-release tablet: 1/day	

Abbreviations: ACE—angiotensin-converting enzyme; ARB—AngII receptor blockers; d—dose; ADR—adverse drug reaction; DM—diabetes mellitus; CCB—calcium channel blockers.

## Data Availability

Not applicable.

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
