# Peer review of "Hypertension in Patients with Insulin Resistance: Etiopathogenesis and Management in Children"

_ijms, 2022, doi:10.3390/ijms23105814_

Round 1

Reviewer 1 Report

Veronica Maria Tagi and collaborators present a review on Hypertension in patients with Insulin Resistance: Etiopathogenesis and Management in Children in where they address how insulin resistance leads to systemic hypertension in children, the approach to the subject is clear, however I have some suggestions that could improve this manuscript.

Resultados de traducción

In the abstract, various abbreviations are required, for example, insulin resistance can be abbreviated as IR, hypertension as HS (systemic hypertension), diabetes as DM (diabetes mellitus). These remarks on the abbreviations should be applied throughout the text. e.g., insulin resistance IR lines; 22,27,44, 49,63,68,71,72,96,98,157,180,187,258,273,295,328,357 to name a few. in the same sense add an abbreviation for systemic hypertension for example hypertension (systemic HS) in the lines; 22,52,55,58,63,64,68,69,118,272,283. In the same sense add an abbreviation for diabetes mellitus for example lines; 55,59,60,65,92, 118,143,134,159,206,229, In line 324 mention is made of angiotensin II and the abbreviation is added. However, later when the mention of angiotensin II appears again, the abbreviation is no longer used, for example line 191. In line 31 the abbreviation NO appears for the first time, but it is not developed, please develop this NO = nitric oxide. Substitute the word beta-cell for the Greek letter that corresponds to beta. In line 56 appears the abbreviation for CVD, what does it mean? develop please On line 34 the abbreviation for type 2 diabetes (T2D) appears but on the bottom line 35 you no longer use it, because. The same example happens with blood pressure (BP) line 24 but then you no longer use it, for example line 55, 286, 332. In line 44, mention is made of the HOMA index. Please add that it is a marker to assess insulin resistance and add among which parameters in this index it is already considered insulin resistance. in line 80 replace plasma membrane with cell membrane. Page 2 and end of paragraph 2 beginning of paragraph 3 it is mentioned how decreased hyperinsulinemia contributes to vascular resistance, although this is true, the decrease in nitric oxide is the result of the uncoupling of endothelial nitric oxide synthase by hyperinsulinemia and insulin resistance. please add and develop how this inhibition is carried. line 113 which is HTN. line 123 please add a colon between ESH and BP. in lines 122 and 124 the abbreviation for AAP appears but in line 126 it appears as AAPs, please homologate. on lines 202 the abbreviation for ARB appears but on line 214 it appears as ARBs, and on page 7 line 227 there is no abbreviation for this, please homologate. The same example appears for the header of table 1 for ESHs but in the table it appears as ESH, please homologate. page 7, paragraph 8 line 263, add abbreviation (RAAS) page 4 line 142 please change.... page 4 paragraph 170 please restructure the opening sentence paragraph 8 because it is not understood. page 4 paragraph 8, line 175 remove prostaglandin 2 because it is a vasoconstrictor molecule and not a vasodilator like prostaglandin 1. Page 5 paragraph 3, line 195 develop the RAS abbreviation and standardize the same abbreviation of line 263. page 8, paragraph 5 line 309, which is CV please develop. Page 2, paragraph 6 line 83 you mention figure 1 but the figure does not exist in the manuscript. Please improve the cell spacing in tables because the information is very crowded. the units in a table are placed at the beginning if it is in percentiles, since it is no longer necessary to repeat the same unit in each of the values, this is not presentable, please fix the tables. also eliminate by dose, within the table, if you put at the beginning that it is per dose, there is no need to mention it again in the entire table, the same thing happens with the dose interval, eliminate "day", which means per d, I think you mean per day please homologate but as I already mentioned this can be eliminated.

Author Response

Dear reviewer 1,

Thank you for your comments and suggestions. As required:

  • In our revised work we have added and edited all the abbreviations required.
  • HOMA index and its normal values have been illustrated in lines 43-45.
  • The inhibition of nitric oxide production has been better explicated in lines 77-79.
  • Figure 1 has been added.
  • Tables have been modified as suggested.

Reviewer 2 Report

I enjoyed reviewing this interesting manuscript. The paper is well written and updated. This reviewer only raises a few comments.

1- It would be useful for the readers if the authors add a figure describing the pathophysiological mechanisms linking IR and hypertension in children.

2- Metabolic syndrome (MS) and non-alcoholic fatty liver disease (NAFLD) are two different entities sharing common clinical and physio-pathological features, with insulin resistance as the most relevant (Antioxidants (Basel). 2021 Feb 10;10(2):270. doi: 10.3390/antiox10020270). The increased MS incidence worldwide, above all due to changes in diet and lifestyle, is associated with an equally significant increase in NAFLD, in children too. This issue and above reference should be commented on in the manuscript.

3- Hypertension is often associated with other risk factors for IR present in metabolic syndrome, particularly type 2 diabetes, in both adults and children. In adults, through randomized multicenter studies, it has recently been possible to demonstrate the importance of intensified multifactorial treatment to reduce morbidity in such situations (Cardiovasc Diabetol, 2021, 20:145. doi: 10.1186/s12933-021-01343-1). It would be desirable that RCTs were designed to evaluate the long-term clinical impact of controlling the risk factors associated with IR also in the pediatric population.

Author Response

Dear reviewer 2,

Thank you for your comments and suggestions. In our revised work, as required:

  1. We added a figure describing the pathophysiological mechanisms linking IR and hypertension in children:
  2. We mentioned NAFLD citing the suggested paper;
  3. We mentioned the fact that in adults it has recently been possible to demonstrate the importance of intensified multifactorial treatment to reduce morbidity in such situations, citing the suggested paper.

Round 2

Reviewer 1 Report

Dear editor and authors of the review article, although almost all the corrections and amendments made to the first version of the manuscript have already been satisfactorily addressed and corrected, I would not like to confirm my decision yet, since I believe that both tables can still be improved , so that you can have greater understanding for the reader. I sent you an example that could serve as a guide to have what you think for both tables. Also figure 1 mentioned in the manuscript still does not appear or is it found in the supplementary material?...... Without further ado for the moment I say goodbye to you.
postscript the example of the table is only a suggestion for your understanding thanks

ESH

AAP

Category

0-15 years

SBP and/or DBP (Percentile)

≥ 16 years SBP and/or DBP

(Values in Percentiles and mmHg)

1-12 years

(Percentiles values)

≥ 13 years

(mmHg)

Normal

<90th

<130/85

<90th

<120/80

High-Normal/

Elevated BP

≥90th to <95th

130-139/85-89

≥90th to 95th

120/<80 to129/<80

Stage 1 HS

95th to 99th + 5 mmHg

140-159/90-99

95th to 95 +12 mmHg

130/80 to 139/99

Stage 2 HS

>99th + 5 mmHg

160-179/100-109

≥95th + 5 mmHg

≥140-90

Abbreviation: ESH=????, AAP=?????, HS=?????, BP=??????, th=??????

Author Response

Dear reviewer 1,

Thank you again for your comments and suggestions, which we highly appreciated.

  • Regarding the two tables, we have further modified them according to your indications. We hope that the current version corresponds to your idea of ​​a table understandable to readers.
  • Regarding figure 1, we had uploaded it as a separated file in the previous version, but now we have put it directly into the main text, in order to make it visible to you.
